

**Variation in altitude of high-frequency enhanced plasma line by the pump near**
**the 5th electron gyro-harmonic**
Jun Wu[a*], Jian Wu[a], Michael. T. Rietveld[b] , Ingemar. Haggstrom[c] , Haisheng Zhao[a],
Tong Xu[a], Zhengwen Xu[a]
[a]*National Key laboratory of electromagnetic environment, China research institute of*
*radio wave propagation, Beijing, 102206, China*
[b]*EISCAT Scientific Association, 9027 Ramfjordbotn, Norway*
[c]*EISCAT Scientific Association, SE-981 92 Kiruna, Sweden*
**Abstract**
During an ionospheric heating campaign carried out at the European Incoherent
Scatter Scientific Association (EISCAT), the ultra high frequency incoherent scatter
(IS) radar observed a systematic variation in the altitude of the high-frequency
enhanced plasma line (HFPL), which behaves depending on the pump frequency.
Specifically, the HFPL altitude becomes lower when the pump lies above the 5th
gyro-harmonic. The analysis shows that the enhanced electron temperature plays a
decisive role in the descent in the HFPL altitude. That is, on the traveling path of the
enhanced Langmuir wave, the enhanced electron temperature can only be matched by
the low electron density at a lower altitude so that the Bragg condition can be satisfied,
as expected from the dispersion relation of Langmuir wave.
**Keywords:** ionospheric heating, incoherent scatter radar, enhanced plasma line,
altitude, Bragg condition.



## 1. Introduction

The oscillation two stream instability (OTSI) and the parametric decay instability (PDI) have been extensively investigated [Silin 1965; DuBois and Goldman 1965, 1967; Perkins and Flick 1971; Rosenbluth 1972; Drake et al.,, 1974; Perkins, et al.,, 1974; Kuo and Cheo 1978; Wu et al.,, 2006; Wu et al.,, 2007). As the signatures of the PDI and OTSI, the high-frequency enhanced plasma line (HFPL) and the high-frequency enhanced ion line (HFIL) are observed by the incoherent scattering (IS) radar during the ionospheric heating campaign. Using those observations of IS radar, the IS spectrum (Kuo and Fejer, 1972; Stubbe et al., 1992; Kohl et al., 1993; Carlson et al., 1972; Gordon and Carlson,1974; Kantor, 1974; Hagfors et al., 1983; Dubois et al., 1988; Nordling et al., 1988 ; Stubbe et al., 1985), the pump threshold for the PDI and OTSI (Fejer, 1979; Bezzerides and Weinstock, 1972; Weinstock and Bezzerides,   1972), the temporal properties of the PDI and OTSI (Kohl et al., 1993; Gordon and Carlson, 1974; Kantor, 1974; Stubbe et al., 1985; Carlson et al., 1972; Jones et al., 1986) and the altitude properties of the HFPL and HFIL (Stubbe et al., 1992 ; Kohl et al., 1987, 1993; Djuth et al., 1994; Ashrafi et al., 2006; Wu et al., 2017a, 2018b) were examined.

The enhanced Langmuir wave and ion acoustic wave are usually excited in the altitude range from the reflection altitude of the pump to the altitude where the heavy Landau effect on Langmuir wave may take place (Stubbe et al., 1992). However, the enhanced Langmuir wave and ion acoustic wave can't be observed by IS radar in the exciting altitude range, but at an altitude where the Bragg condition is satisfied



(Stubbe et al., 1992; Kohl et al., 1987, 1993). Some usual observations of the ultra
high frequency (UHF) radar at European Incoherent Scatter Scientific Association
(EISCAT) show that the HFIL altitude is about ~ 3 km – ~ 5 km higher than the HFPL
altitude (Stubbe et al., 1992; Kohl et al., 1993). Additionally, the altitude extending of
~ 3 km – ~ 5 km frequently appears in the power profile of the HPIL, but does not in
the power profile of the HFPL (Stubbe et al., 1992; Kohl et al., 1993). Moreover, some
observations at EISCAT illustrated that a descent in the altitude of the plasma
turbulence took place over tens of seconds after the pump on, which was most likely
attributed to the modification in electron density by the ionospheric heating (Djuth et
al., 1994). UHF radar at EISCAT observed the descent in the HFIL altitude from ~
230 km to ~ 220 km within ~ 60 s, which was also attributed to the modification in
electron density (Ashrafi et al., 2006).

Although those variations in the HFPL and HFIL altitudes were attributed to the

enhanced electron temperature and the modified electron density, the dominant one of
which was not clearly identified (Wu et al., 2017a). Furthermore, it was identified that
the enhanced electron temperature dominated over the modified electron density in
the variation in the HFIL altitude (Wu et al., 2018b). As a further work, this paper
examines the variation in the HFPL altitude in more detail. Indeed, the dispersion
behavior of Langmuir wave is very different from that of ion acoustic wave.
**2. Experiment and data**

An ionospheric heating campaign was performed at EISCAT at 12:32:30 UT –

14:30 UT (universal time) on Mar. 11, 2014. The experiment arrangement has been





described in more detail by Wu et al., (2016, 2017b). Briefly, the EISCAT heater
(Rietveld et al., 1993, 2016) radiated the O mode pump in the frequency band of 6.7
MHz – 7 MHz, and the UHF IS radar was operated as the leading diagnostic means.
The pump frequency $f_{HF}$ was stepped down and up in a step of 2.804 kHz with a
period of 10 s as shown in those bottom panels in **Figure 1, Figure 2** and **Figure 3**.
During the experiment, the local geomagnetic was relatively quiet. At an altitude of
200 km, the total geomagnetic varied in the range of 49202 nT – 49233 nT.

Considering the variation in the intensity of ion line, we adopt a convention for

the following discussion: the $f_{HF}$ band of 6.7 MHz – 7 MHz can be divided into
three daughter bands, that is, the higher band (HB, above $5f_{ce}$), the gyro-harmonic
band (GB, very close to $5f_{ce}$) and lower band (LB, below $5f_{ce}$), where $f_{ce}$
represents the electron gyro-frequency (Wu et al., 2016, 2017a, 2017b, 2018a, 2018b,
2019). For instance, in the 1st heating cycle, the HB is set as 7 MHz – ~ 6.871028
MHz, the GB as ~ 6.868224 MHz – ~ 6.837383 MHz and the LB as ~ 6.834579
MHz – 6.7 MHz, which temporally correspond to the time intervals of 12:30:00 UT
– 12:37:40 UT, 12:37:50 UT – 12:39:40 UT and 12:39:50 UT – 12:48:00 UT,
respectively. Actually, the frequency division in each heating cycle should be
somewhat different from each other due to the slight disturbance of the geomagnetic.

From the 1st panel to the 6th panel in **Figure 1**, the normalized plasma lines at

those altitudes of 210.25 km, 207.32 km, 204.39 km, 201.45 km, 198.52 km and
195.58 km are successively given, which lie in the frequency range of -6.7 MHz –
–7.25 MHz. One can find that those HFPLs in the GB and HB lie at frequency





$f_{HF} - 9.45$ kHz as the expected decay line from the PDI. In the GB, those strong
HFPLs of up to ~ 1 occur at an altitude of 201.45 km in the 1st heating cycle, at an
altitude of 210.25 km in the 2nd heating cycle and at an altitude of 207.32 km in the
3rd and 4th heating cycles respectively. In the HB, however, those strong HFPLs of
up to ~ 1 descend in altitude, that is, they are located at an altitude of 198.52 km in the
1st heating cycle, at altitudes of 207.32 km and 204.39 km in the 2nd heating cycle, at
an altitude of 204.39 km in the 3rd and 4th heating cycles. On the other hand, in the
LB, the HFPL has not appeared at any of those altitudes due to the absence of the PDI
and OTSI (Wu et al., 2019).

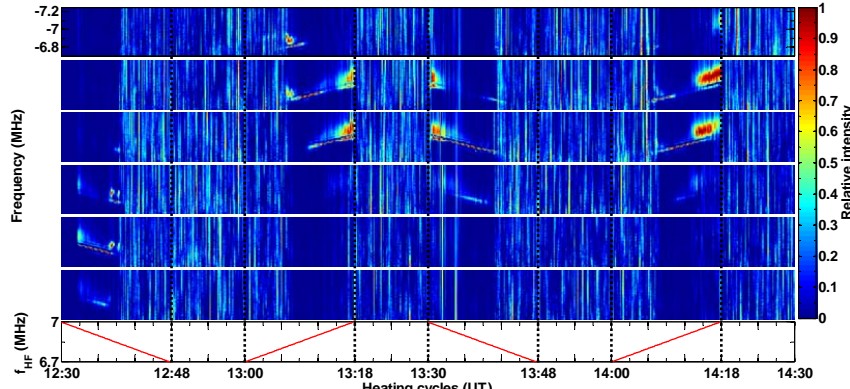


**Figure 1.** The plasma lines versus $f_{HF}$ (the heating cycles), where the 1st panel is for
an altitude of 210.25 km, the 2nd panel for 207.32 km , the 3rd panel for 204.39 km,
the 4th panel for 201.45 km, the 5th panel for 198.52 km, the 6th panel for 195.58 km
and the 7th panel for $f_{HF}$ (the heating cycles), successively from top to bottom.
**Figure 2** gives the altitude profile of $T_e/T_{e0}$ as a function of $f_{HF}$, where $T_e$
is the electron temperature and $T_{e0}$ the undisturbed electron temperature. At an
altitude of ~ 200 km, $T_e/T_{e0}$ immediately enhances when heating on, and obviously
varies with $f_{HF}$. $T_e/T_{e0}$ strongly enhances up to ~ 1.5 in the LB, whereas it slightly



enhances up to ~ 1.25 in the HB. In the GB, $T_e/T_{e0}$ approximately reach ~ 1.2.
Evidently, $\left(T_e/T_{e0}\right)_{LB} > \left(T_e/T_{e0}\right)_{HB} > \left(T_e/T_{e0}\right)_{GB}$, where $\left(T_e/T_{e0}\right)_{LB}$, $\left(T_e/T_{e0}\right)_{HB}$ and
$\left(T_e/T_{e0}\right)_{GB}$ represent $T_e/T_{e0}$ in the LB, HB and GB respectively. This variation in
$T_e/T_{e0}$ depends on the dispersion behavior of the excited upper hybrid waves at the
upper hybrid altitude (Wu et al., 2017b), where is ~ 2 km – ~ 10 km below the
reflection altitude of the pump (Gurevich, 2007).

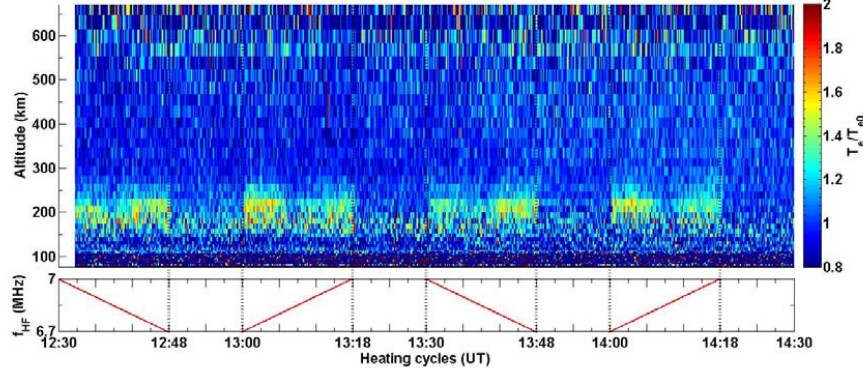


**Figure 2.** $T_e/T_{e0}$ versus $f_{HF}$ (the heating cycles), where $T_{e0}$ is obtained by
averaging the electron temperature over the final 5 minutes of the UHF radar
observations at 14:25 UT – 14:30 UT.
**Figure 3** is the altitude profile of $N_e/N_{e0}$ as a function of $f_{HF}$, where $N_e$ is
the electron density and $N_{e0}$ the undisturbed electron density. In the 3rd and 4th
heating cycles, the enhanced $N_e/N_{e0}$ of up to ~ 1.4 take place in the GB and HB and
can be seen in a narrow region around an altitude of ~ 200 km. In accordance with the
standard IS analysis, the enhanced $N_e/N_{e0}$ should not correspond to the real
enhancement in the electron density, but to the HFIL excited by the PDI and OTSI (Wu
et al., 2017b). On the other hand, no apparent enhancement in $N_e/N_{e0}$ takes place
around an altitude of ~ 200 km in the 1st and 2nd heating cycles. This may be due to





the high background electron density and the ambiguity of radar measurement (Wu et
al., 2017b). Additionally, the enhanced $N_e/N_{e0}$ appears over a wide altitude range of
~ 250 km – ~ 670 km, which is hardly explained by the standard IS analysis and is
open (Wu et al., 2017b).

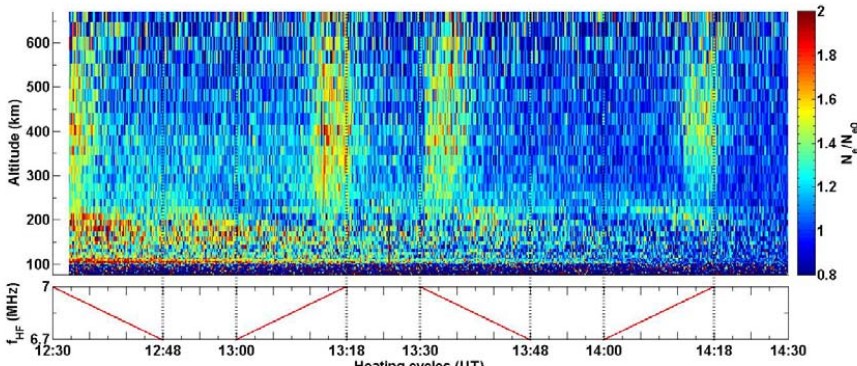


**Figure 3.** $N_e/N_{e0}$ versus $f_{HF}$ (the heating cycles), where $N_{e0}$ is obtained by
averaging the electron density over the final 5 minutes of the UHF radar observations
at 14:25 UT – 14:30 UT.
In summary, **Figure 1** shows that (1) the HFPL altitude in the 1st heating cycle is
far lower than that in the 2nd, 3rd and 4th heating cycles; (2) interestingly enough,
those HFPL altitudes in the GB and HB systematically vary with $f_{HF}$, that is, the
HFPL altitude in the HB is slightly lower than that in the GB. Additionally, **Figure 2**
implies that $T_e$ also systematically varies with $f_{HF}$, whereas $N_e$ does not as
illustrated in **Figure 3**.
**3. Discussion**
OTSI and PDI can be excited in the altitude range of (Stubbe et al., 1992)
$$h_0 - 0.1H \leq h_{ex} < h_0 \qquad (1)$$
where $h_0$ is the reflection altitude of the pump, $H$ is the scale altitude and $h_{ex}$ is





the exciting altitude of the PDI and OTSI. For a typical ionosphere, due to the
monotonous change in the profile of $N_e$ below the ionospheric peak, $h_{ex}$ in the HB
should be higher than that in the GB. In **Figure 3**, it is evident that $N_e / N_{e0}$ in the 1st
heating cycle reaches ~ 1.7 near an altitude of 200 km and is far larger than that in the
2nd, 3rd and 4th heating cycles. This implies that $h_0$ in the 1st heating cycle should
be far lower than that in the 2nd, 3rd and 4th heating cycles. Correspondingly, $h_{ex}$
and the HFPL altitude in the 1st heating cycle should be far lower than that in the 2nd,
3rd and 4th heating cycles.

However, function (1) fails to explain that the HFPL altitude in the HB is slightly

lower than that in the GB. Considering an field-aligned and monostatic operating
observation, the enhanced Langmuir wave traveling in a non-uniform and stationary
ionosphere should satisfy the dispersion relation (Kohl et al., 1993)

$$\omega_L^2 = \omega_{pe}^2 + \gamma \frac{K_B T_e}{m_e} k_L^2 \qquad (2)$$

where $\omega_L$ is the angular frequency of Langmuir wave, $\omega_{pe}$ is the Langmuir angular
frequency of ionospheric plasma, $\gamma$ is the adiabatic index, $K_B$ is the Boltzmann
constant, $k_L$ is the wave number of Langmuir wave, and $m_e$ is the electron mass.

When the enhanced Langmuir wave travels in a non-uniform and stationary

ionosphere, $k_L$ may change, whereas $\omega_L$ will not change. That is, $k_L$ should
depend on $\omega_{pe}$ and $T_e$ on the traveling path of the enhanced Langmuir wave, as
expected from function (2). This implies that at a particular altitude, $k_L$ will satisfies
the Bragg condition, namely, $k_L = 2k_r$, and the enhanced Langmuir wave should be
observed by radar, where $k_r$ is the wave number of radar. Then, considering $T_e = T_e'$,



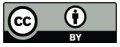

the enhanced Langmuir wave should be observed at an altitude of $h'$ where
$2k_r = k_L = \sqrt{\dfrac{(\omega_L^2 - \omega_{pe}'^2)m_e}{\gamma K_B T_e'}}$, which is a derivation of function (2). On the other hand,
$T_e = T_e''$ is considered, then the enhanced Langmuir wave should be observed at other
altitude of $h''$, where $2k_r = k_L = \sqrt{\dfrac{(\omega_L^2 - \omega_{pe}''^2)m_e}{\gamma K_B T_e''}}$. Obviously, $\dfrac{\omega_L^2 - \omega_{pe}''^2}{T_e''} = \dfrac{\omega_L^2 - \omega_{pe}'^2}{T_e'}$
can be obtained. Furthermore, if $T_e'' > T_e'$, then $\omega_{pe}'' < \omega_{pe}'$. Due to the monotonous
profile of $\omega_{pe}$ below the ionospheric peak, $h'' < h'$ will be obtained. In other word,
on the traveling path of the enhanced Langmuir wave, the higher $T_e$ is, the lower the
observing altitude of the enhanced Langmuir wave is. The fact is
$(T_e/T_{e0})_{HB} > (T_e/T_{e0})_{GB}$ as shown in Figure 2. As a result, the HFPL altitude in the
HB should be lower than that in the GB as shown in Figure 1.

As an example, the HFPL in the 4th heating cycle is examined. The left panel of

**Figure 4** respectively gives the profiles of $\omega_L^2 - \omega_{pe}^2$, $T_{eGB}$ and $T_{eHB}$ in the altitude
range of 190 km – 230 km in the 4th heating cycle. Here, the profile of $\omega_L^2 - \omega_{pe}^2$ is
not distinguished in the GB and HB, implying an assumption that the profiles of $N_e$
was not modified by the ionospheric heating. Indeed, it is difficult to measure the
slight modification in electron density due to (1) $N_e$ is much variable in space and
time, and (2) the artificial modification in $N_e$ is relatively small (Rietveld et al.,
2003). Also, **Figure 3** really exhibits that no real modification in $N_e/N_{e0}$ is induced
by the ionospheric heating in the altitude range examined (Wu et al., 2017b).
Obviously, $\omega_L^2 - \omega_{pe}^2$ monotonically decreases with the ascent in altitude and has a
vertical gradient of $\sim -3.1 \times 10^{15}$ rad$^2$s$^2$km$^{-1}$ in the altitude range of 200 km – 230





km. Moreover, $\omega_L^2 - \omega_{pe}^2$ becomes negative above an altitude of ~ 208.5 km due to
the increasing $N_e$ with the ascent in altitude. In addition, the profile of $T_{eGB}$
demonstrates the gradient of ~ 6.09 Kkm$^{-1}$ within the altitude range of 190 km –
200 km and ~ – 8.85 Kkm$^{-1}$ within the altitude range of 200 km – 230 km,
whereas $T_{eHB}$ demonstrates the gradient of ~ 40.82 Kkm$^{-1}$ within the altitude range
of 190 km – 200 km and ~ – 17.77 Kkm$^{-1}$ within the altitude range of 200 km –
230 km. This implies that the strongest enhancement in $T_e$ takes place an altitude of
~ 200 km and the thermal energy should be conducted along the magnetic field within
an extending altitude range.

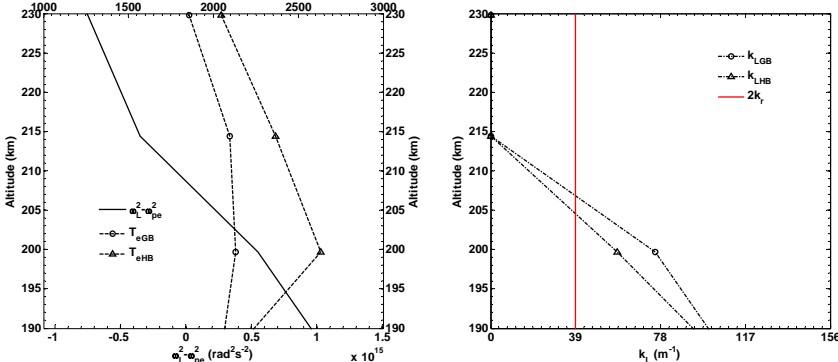


**Figure 4.** The altitude profiles of $\omega_L^2 - \omega_{pe}^2$, $T_{eGB}$, $T_{eHB}$ (the left panel), $k_{LGB}$, $k_{LHB}$
and $k_r$ (the right panel), where $\omega_L = 2\pi \times 6.8$ MHz, $\omega_{pe} = 2\pi \times 8.9\sqrt{N_e}$, $N_e$ is
obtained by averaging over the time interval of [14:07:20 UT, 14:09:10 UT], $T_{eGB}$

and $T_{eHB}$ are obtained by averaging over the time intervals of [14:07:20 UT,

14:09:10 UT] and [14:11:20 UT, 14:18:00 UT], respectively, and $k_r = 19.5$ m$^{-1}$ is the

wave number of EISCAT UHF radar.

In the right panel of **Figure 4**, the profiles of $k_{LGB}$ and $k_{LHB}$ in the altitude



range of 190 km – 230 km in the 4th heating cycle are demonstrated, where $k_{\mathrm{LGB}}$ and
$k_{\mathrm{LHB}}$ represent the wave numbers of the enhanced Langmuir wave in the GB and HB.
The profile of $k_{\mathrm{LGB}}$ has a gradient of $\sim -2.78\ \mathrm{m}^{-1}\mathrm{km}^{-1}$ within the altitude range of
190 km – 200 km and $\sim -5.1\mathrm{m}^{-1}\mathrm{km}^{-1}$ within the altitude range of 200 km – 214.4
km, and $k_{\mathrm{LGB}} = 2k_{\mathrm{r}}$ takes place at an altitude of ~ 206.8 km. Moreover, the profile of
$k_{\mathrm{LHB}}$ demonstrates a gradient of $\sim -3.15\ \mathrm{m}^{-1}\mathrm{km}^{-1}$ within an altitude range
examined, and $k_{\mathrm{LHB}} = 2k_{\mathrm{r}}$ at an altitude of ~ 204.5 km. This indicates that the
enhanced $T_{\mathrm{e}}$ on the traveling path can remarkably impact on $k_{\mathrm{L}}$, and the enhanced
Langmuir waves in the GB and HB should be observed at different altitude, namely, ~
206.8 km in the GB and ~ 204.5 km in the HB respectively. Thus, the altitude
difference between the HFPL altitudes in the GB and HB is 2.3 km as illustrated in the
right panel of **Figure 4**. Taking the height resolution of ~ 3 km of EISCAT UHF radar
into account, the HFPL altitudes in the GB and HB in the 4th heating cycle shown in
**Figure 1** are in perfect agreement with the altitudes of $k_{\mathrm{LGB}} = 2k_{\mathrm{r}}$ and $k_{\mathrm{LHB}} = 2k_{\mathrm{r}}$
illustrated in the right panel of **Figure 4**. In addition, $k_{\mathrm{LGB}}$ and $k_{\mathrm{LHB}}$ become zero
above an altitude of ~ 208.5, implying the enhanced Langmuir wave will be reflected
at an altitude of ~ 208.5 km.
Usually, $\omega_{\mathrm{pe}}$ is on the order of $10^{6}$ and $\sqrt{\gamma \dfrac{K_{\mathrm{B}}T_{\mathrm{e}}}{m_{\mathrm{e}}}}k_{\mathrm{L}}$ is on the order of $10^{3}$
for a typical ionosphere, implying that $N_{\mathrm{e}}$ dominates over $T_{\mathrm{e}}$ in $k_{\mathrm{L}}$. However, this
does not imply that the enhanced $T_{\mathrm{e}}$ is independent of the HFPL altitude. Indeed, on
the traveling path of Langmuir wave, an remarkable enhancement in electron
temperature owing to an ionospheric heating will take significant impact on $k_{\mathrm{L}}$. For a

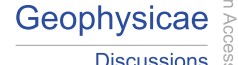

225 large gradient profile of $N_e$, an somewhat enhancement in $T_e$ may lead to an

226 remarkable descent in the HFPL altitude. Moreover, if a small gradient profile of $N_e$

227 is considered, that is, $N_e$ can be approximately considered as a constant, then $k_L$

228 will be mainly determined by the profile of $T_e$.

229 **4. Conclusions**

230  A systematic variation in the HFPL altitude induced by the pump near the 5th

231 gyro-harmonic at EISCAT, is paid attention. The IS radar observation demonstrates

232 that the HFPL altitude and the electron temperature behave as a function of the pump

233 frequency. More specifically, when the pump frequency approaches the 5th

234 gyro-harmonic from below, the electron temperature is somewhat enhanced, and the

235 HFPL is observed at an altitude as expected. When the pump frequency sweeps above

236 the 5th gyro-harmonic, however, the electron temperature is prominently enhanced,

237 and the HFPL altitude slightly plunge downward.

238  In conclusion, the HFPL altitude is dependent on the dispersion behavior of the

239 enhanced Langmuir wave and the Bragg condition, and is determined by the profiles

240 of the electron density and the enhanced electron temperature. When heating above

241 the 5th gyro-harmonic, the HFPL altitude plunge downward owing to the thermal

242 effect of ionospheric heating on the traveling path of the enhanced Langmuir wave. In

243 other word, when the pump sweeps above the 5th gyro-harmonic, the IS radar should

244 observe the enhanced Langmuir wave at an lower altitude, where the low electron

245 density can compensate the remarkably enhanced electron temperature so that the

246 Bragg condition can be satisfied, as expected by the dispersion relation of Langmuir



wave.
**Acknowledgments**
We would like to thank the engineers of EISCAT in Tromsø for keeping the
facility in excellent working condition and Tromsø Geophysical Observatory, UiT The
Arctic University of Norway, for providing the magnetic data of Tromsø recorded on
11 Mar. 2014. The data of UHF radar can be obtained freely from EISCAT
(http://www.eiscat.se/schedule/schedule.cgi). The EISCAT Scientific Association is
supported by China (China Research Institute of Radiowave Propagation), Finland
(Suomen Akatemia of Finland), Japan (the National Institute of Polar Research of
Japan and Institute for Space-Earth Environmental Research at Nagoya University),
Norway (Norges Forkningsrad of Norway), Sweden (the Swedish Research Council)
and the UK (the Natural Environment Research Council).

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
