# Peer review of "1. Introduction"

_Annales Geophysicae, 2019_

## Referee Comment (RC1) · Anonymous Referee #1 · 12 Apr 2019

The manuscript describes experimental results from EISCAT UHF radar observations in the course of HF pumping experiment at Tromsø near the fifth electron gyrofrequency on 11 March 2014 from 12.30 – 14.30 UT. The particular attention is paid to the observing altitude of the HF-enhanced plasma lines excited by an O mode pump in the course of the frequency stepping through the fifth electron gyrofrequency. Authors concluded that EISCAT UHF radar observations have demonstrated that the enhanced electron temperature plays a decisive role in the descent in the HFPL altitude. This manuscript repeats the results which have been yet published by authors for the same experiment on 11 March 2014 and for the same time interval from 12.30 – 14.30 UT in following articles:

[Figure]

Wu J., Wu J., Rietveld M. T., Haggstrom I., Xu Z., Zhang Y., Xu T., Zhao H. The Intensities of High Frequency-Enhanced Plasma and Ion Lines During Ionospheric Heating. JGR Space Physics. 124(1). P.603-615. doi:10.1029/2018JA025918, 2018. Wu, J., Wu J., Rietveld M. T., Haggstrom I., Xu Z., Zhao H. The extending of observing altitudes of plasma and ion lines during ionospheric heating. Journal of Geophysical Research: Space Physics, 123(1), 918-930, doi.org/10.1002/2017JA024809 2018 Wu, J., J. Wu, M. T. Rietveld, I. Haggstrom, H. Zhao, and Z. Xu, The behavior of electron density and temperature during ionospheric heating near the fifth electron gyrofrequency, J. Geophys. Res. Space Physics, 122, doi:10.1002/2016JA023121, 2017. Wu J., J. Wu, M T Rietveld, I Haggstrom, H. Zhao and Z. Xu, Altitude and intensity characteristics of parametric instability excited by an HF pump wave near the fifth electron harmonic, Plasma Sci. Technol., 19(12), 2017. Wu J, J. Wu, H. Zhao and Z. Xu, Analysis of incoherent scatter during ionospheric heating near the fifth electron gyrofrequency, Plasma Sci. Technol., 19(4), doi:10.1088/2058-6272/aa58db, 2017.

Note, that the only experiment on 11 March 2014 is not unique itself. There were a lot of other frequency stepping experiments near the electron gyroharmonics at EISCAT. For a example, Borisova et al (Radiophys. & Quantum Electron., 2016, 58, 8, 561-585) has described and analyzed a series of EISCAT O-mode HF pump frequency stepping experiments near the fifth electron gyroharmonic carried out on 22, 23, 25 and 26 October 2013, when the features and behaviors of HF-enhanced ion and plasma lines from EISCAT UHF radar observations were considered in the combination with the artificial field-aligned irregularities from the CUTLASS (SuperDARN) observations and spectral features of the stimulated electromagnetic emission measurements.

The important comment is also that the Discussion section, based on the articles by Stubbe et al., 1992; Djuth et al., 1994, is appropriative only for the vertical incident angles. However, the experiment on 14 March 2014 was conducted under HF pumping towards the magnetic zenith. O-mode HF pump waves at the magnetic zenith reflect below the standard reflection layer at vertical incident angles (see Mishin et al., JGR,

2004, V.109, A02305; Ann. Geophys., 2005, 23, p.47-53 and Fig.21 from Gurevich, 2007). Moreover, at high pump frequencies near the 5th electron gyro-harmonic (fH $\sim$ 5fce) the pump wave is reflected near the upper hybrid resonance altitude. Therefore, the careful estimations of the reflection altitudes at MZ pumping are necessary.

It should also be noted that the analysis of changes in the excited height of PDI and OTSI instabilities, taking into account the dispersion relations, is carried out under the assumption that the parameters of the ionospheric plasma are monotonous. However the spatial changes of the electron density and temperature versus fHF (the time of heating cycles) are more complicated that are not taken into account in the analysis of pump frequency variations around the fifth electron gyro-harmonic.

Conclusion. The manuscript adds nothing to the results, which were already published by authors. The Discussion section is not correct for the conditions of the experiment on 14 March 2014, when HF pumping was produced towards the magnetic zenith and not considered the behavior and features of the parameters of the ionosphere near electronic gyroresonances.. I cannot recommend the manuscript for the publication.

---

## Referee Comment (RC2) · Anonymous Referee #2 · 12 Apr 2019

Review of "Variation in altitude of high-frequency enhanced plasma line by the pump near the 5th electron gyro-harmonic" by Wu et Al.

This manuscript presents experimental results of pumping the ionospheric plasma by transmitting a powerful radio wave into the ionosphere from the EISCAT Heating facility and studying the plasma response with the EISCAT UHF incoherent scatter radar. Interpretations of the experimental results are given in terms linear dispersion properties of Langmuir waves in unmagnetized plasma.

It is not clear what is new in the manuscript. Dependencies of the HFPL and Langmuir dispersion characteristics on electron density and temperature as well as pump fre-
quency are well known since decades. Also, electron gyroharmonic results where, for example, discussed already by Honary et al. (JGR 100, 21489, 1995). Further, several papers have been published by the authors from the same two hours of experiments, even with the same figures(!), but no information is given on how these papers relate to each other and to the present treatment. In addition, the Discussion is confusing and appears logically inconsistent. For example, the paragraph lines 161-177 concerns a logic that is applicable for a constant Langmuir wave frequency, that is constant pump frequency. But in the experiments the pump frequency is changed. Both the pump frequency and $T_e$ influence the height at which Langmuir waves are detected by the radar. These two should be kept separate, not mixed together. Taken together, therefore, I cannot recommend publication of this manuscript.

---

## Author Comment (AC1) · 14 Apr 2019

**Reviewer 1**

Thank you for the comments and suggestions. The comments given by reviewer are listed in black

and ours replies are listed in red as below:

**Comments to the Author**

1. This manuscript repeats the results which have been yet published by authors for the same experiment on 11 March 2014 and for the same time interval from 12.30 - 14.30 UT in following articles:

(1) Wu J., Wu J., Rietveld M. T., Haggstrom I., Xu Z., Zhang Y., Xu T., Zhao H. The Intensities of High Frequency-Enhanced Plasma and Ion Lines During Ionospheric Heating. JGR Space Physics. 124(1). P.603-615. doi:10.1029/2018JA025918, 2018.

(2) Wu, J., Wu J., Rietveld M. T., Haggstrom I., Xu Z., Zhao H. The extending of observing altitudes of plasma and ion lines during ionospheric heating. Journal of Geophysical Research: Space Physics, 123(1), 918-930, doi.org/10.1002/2017JA024809 2018

(3) Wu, J., J. Wu, M. T. Rietveld, I. Haggstrom, H. Zhao, and Z. Xu, The behavior of electron density and temperature during ionospheric heating near the fifth electron gyrofrequency, J. Geophys. Res. Space Physics, 122, doi:10.1002/2016JA023121, 2017.

(4) Wu J., J. Wu, M T Rietveld, I Haggstrom, H. Zhao and Z. Xu, Altitude and intensity characteristics of parametric instability excited by an HF pump wave near the fifth electron harmonic, Plasma Sci. Technol., 19(12), 2017.

(5)Wu J, J. Wu, H. Zhao and Z. Xu, Analysis of incoherent scatter during ionospheric heating near the fifth electron gyrofrequency, Plasma Sci.Technol., 19(4), doi:10.1088/2058-6272/aa58db, 2017.

The enhanced electron temperature as a function of pump frequency

**Fig 1. the enhanced electron temperature as a function of pump frequency**

The fact is that from the experimental observations performed in the interval of 12.30 - 14.30 UT on 11 March 2014, we find **six** interesting phenomenon (scientific questions), namely,

(1) the enhanced electron temperature as a function of pump frequency as shown in Fig 1. This result was reported in published paper [*Wu*, *J.*, *J. Wu*, *M. T. Rietveld*, *I. Haggstrom*, *H. Zhao*, and Z. Xu, The behavior of electron density and temperature during ionospheric heating near the fifth electron gyrofrequency, J. Geophys. Res. Space Physics, 122, doi:10.1002/2016JA023121, 2017; Wu J, J. Wu, H. Zhao and Z. Xu, Analysis of incoherent scatter during ionospheric heating near the fifth electron gyrofrequency, Plasma Sci.Technol., 19(4), doi:10.1088/2058-6272/aa58db, 2017.].

---

## Author Comment (AC2) · 14 Apr 2019

Reviewer 2

Thank you for the comments and suggestions. The comments given by reviewer are listed in black and ours replies are listed in red as below:

Comments to the Author

1. It is not clear what is new in the manuscript. Dependencies of the HFPL and Langmuir dispersion characteristics on electron density and temperature as well as pump frequency are well known since decades. Also, electron gyroharmonic results where, for example, discussed already by Honary et al. (JGR 100, 21489, 1995). Further, several papers have been published by the authors from the same two hours of experiments, even with the same figures(!), but no information is given on how these papers relate to each other and to the present treatment.

[Figure]

The enhanced electron temperature as a function of pump frequency

**Fig 1.** the enhanced electron temperature as a function of pump frequency

The fact is that from the experimental observations performed in the interval of 12.30 – 14.30 UT on 11 March 2014, we find **six** interesting phenomenon (scientific questions), namely,
(1) **the enhanced electron temperature as a function of pump frequency** as shown in **Fig 1**. This result was reported in published paper [*Wu, J., J. Wu, M. T. Rietveld, I. Haggstrom, H. Zhao, and Z. Xu, The behavior of electron density and temperature during ionospheric heating near the fifth electron gyrofrequency, J. Geophys. Res. Space Physics, 122, doi:10.1002/2016JA023121, 2017; Wu J, J. Wu, H. Zhao and Z. Xu, Analysis of incoherent scatter during ionospheric heating near the fifth electron gyrofrequency, Plasma Sci.Technol., 19(4), doi:10.1088/2058-6272/aa58db, 2017.*].

[Figure]

**The extending enhancement in electron density**

**Fig 2.** the altitude extending enhanced electron density

(2) **the altitude extending enhanced electron density** as shown in **Fig 2**. This result was reported in published paper [*Wu, J., J. Wu, M. T. Rietveld, I. Haggstrom, H. Zhao, and Z. Xu, The behavior of electron density and temperature during ionospheric heating near the fifth electron gyrofrequency, J. Geophys. Res. Space Physics, 122, doi:10.1002/2016JA023121, 2017; Wu J, J. Wu, H. Zhao and Z. Xu, Analysis of incoherent scatter during ionospheric heating near the fifth electron gyrofrequency, Plasma Sci.Technol., 19(4), doi:10.1088/2058-6272/aa58db, 2017.*].

[Figure]

**The HFIL in the HB is extending in altitude (green circle), whereas the HFIL in the GB is not (yellow circle).**

**Fig 3.** a remarkable extension of observing altitudes of the HFIL

(3) **a remarkable extension of observing altitudes of the HFIL in the HB** as shown in **Fig 3**. This result was reported in published paper [*Wu, J., Wu J., Rietveld M. T., Haggstrom I., Xu Z., Zhao H. The extending of observing altitudes of plasma and ion lines during ionospheric heating. Journal of Geophysical Research: Space Physics, 123(1), 918-930, doi.org/10.1002/2017JA024809 2018*].

[Figure]

**The intensity of HFIL in the HB is weak (green circle), whereas the intensity of HFIL in the GB is strong (yellow circle).**

**Fig 4.** the variation in the intensity of the HFIL as a function of pump frequency

(4) **the variation in the intensity of the HFIL as a function of pump frequency** as shown in **Fig 4.** This result was reported in published paper [*Wu J., Wu J., Rietveld M. T., Haggstrom I., Xu Z., Zhang Y., Xu T., Zhao H. The Intensities of High Frequency-Enhanced Plasma and Ion Lines During Ionospheric Heating. JGR Space Physics. 124(1). P.603-615. doi:10.1029/2018JA025918, 2018.*]. In addition, as a phase work, we reported the original idea about the intensity of the HFIL in paper [*Altitude and intensity characteristics of parametric instability excited by an HF pump wave near the fifth electron harmonic, Plasma Sci. Technol., 19(12), 2017.* ]

[Figure]

**The HFIL altitude in the HB (green circle) is lower than that in the GB (yellow circle).**

**Fig 5.** a systematic variation in the altitude of the HFIL as a function of pump frequency

(5) **a systematic variation in the altitude of the HFIL as a function of pump frequency**, namely, the altitude of the HFIL in the HB is lower than that in the GB, as shown in **Fig 5.** This result was reported in published paper [*Wu, J., Rietveld, M.T., Häggström, I., Zhao, H., Xu, T. & Xu, Z. (2018). Systematic variation in observing altitude of enhanced ion line by the pump near*

*fifth gyroharmonic. Plasma Science and Technology, 20(12), 125301. https://doi.org/10.1088/2058-6272/aadd44*]. In addition, as a phase work, we reported the original idea about the intensity of the HFIL in paper [*Altitude and intensity characteristics of parametric instability excited by an HF pump wave near the fifth electron harmonic, Plasma Sci. Technol., 19(12), 2017.* ]

(6) **a systematic variation in the altitude of the HFPL as a function of pump frequency**, namely, the altitude of the HFPL in the HB is lower than that in the GB, as shown in **Fig 6.** This result was submitted ANGEO with number angeo-2019-23, namely, the reviewed paper.

[Figure]

The HFPL altitude in the HB (green circle) is slightly lower than that in the GB (yellow circle)

**Fig 6.** a systematic variation in the altitude of the HFPL as a function of pump frequency

As those statements above mentioned, using those observations (data) obtained in the interval of 12.30 – 14.30 UT on 11 March 2014, **six** interesting phenomenon (scientific questions) were studied and published respectively, implying that **although the same observations (data or figures) were used in those published papers, but the focused question in those published papers is very different from each other**. Thus, this manuscript (angeo-2019-23) does **not repeat the study results** which have been published by authors, but only **repeats those radar observations**. Indeed, the heating results at third gyroharmonic have been discussed by Honary et al. (JGR 100, 21489, 1995), which should be cited.

2. In addition, the Discussion is confusing and appears logically inconsistent. For example, the paragraph lines 161-177 concerns a logic that is applicable for a constant Langmuir wave frequency, that is constant pump frequency. But in the experiments the pump frequency is changed. Both the pump frequency and T_e influence the height at which Langmuir waves are detected by the radar. These two should be kept separate, not mixed together. Taken together, therefore, I cannot recommend publication of this manuscript.

This can be due to my poor English. I would like to make some clarity.

When the enhanced Langmuir wave travels in a non-uniform and stationary ionosphere, $k_\mathrm{L}$ may change, whereas $\omega_\mathrm{L}$ will not change. That is, $k_\mathrm{L}$ should depend on $\omega_\mathrm{pe}$ and $T_\mathrm{e}$ on the

traveling path of the enhanced Langmuir wave. This implies that at a particular altitude, $k_L$ will

satisfies the Bragg condition, namely, $k_L = 2k_r$, and the enhanced Langmuir wave should be

observed by radar, where $k_r$ is the wave number of radar.

Although the fact is that $\omega_{LHB}$ is slightly larger than $\omega_{LGB}$, here we consider the enhanced

Langmuir wave at single frequency $\omega_L$ for the sake of simplicity, where $\omega_{LHB}$ and $\omega_{LGB}$ are

respectively the frequency of the enhanced Langmuir wave in the HB and GB.

In the GB, the enhanced Langmuir wave at frequency $\omega_L$ should be observed at an altitude $h_{GB}$

where the Bragg condition is satisfied, namely, $2k_r = k_L = \sqrt{\dfrac{\left(\omega_L^2 - \omega_{peGB}^2\right)m_e}{\gamma K_B T_{eGB}}}$ , where $\omega_{peGB}$

is Langmuir frequency at altitude $h_{GB}$ .

In the HB, the enhanced Langmuir wave at frequency $\omega_L$ should be observed at an altitude $h_{HB}$

where the Bragg condition is satisfied, namely, $2k_r = k_L = \sqrt{\dfrac{\left(\omega_L^2 - \omega_{peHB}^2\right)m_e}{\gamma K_B T_{eHB}}}$ , where $\omega_{peHB}$

is Langmuir frequency at altitude $h_{HB}$ .

Thus, $\sqrt{\dfrac{\left(\omega_L^2 - \omega_{peGB}^2\right)m_e}{\gamma K_B T_{eGB}}} = \sqrt{\dfrac{\left(\omega_L^2 - \omega_{peHB}^2\right)m_e}{\gamma K_B T_{eHB}}}$ should be obtained. In further,

$\dfrac{\omega_L^2 - \omega_{peGB}^2}{T_{eGB}} = \dfrac{\omega_L^2 - \omega_{peHB}^2}{T_{eHB}}$ is obtained. Due to $T_{eGB} < T_{eHB}$ , then

$\left(\omega_L^2 - \omega_{peGB}^2\right) < \left(\omega_L^2 - \omega_{peHB}^2\right)$ . In further, $\omega_{peGB}^2 > \omega_{peHB}^2$ is obtained. Thus, Due to the

monotonous profile of $\omega_{pe}$ below the ionospheric peak, $h_{GB} > h_{HB}$ is obtained.

In other word, with regard to the enhanced Langmuir wave at frequency $\omega_L$ , considering the

dispersion relation $\omega_L^2 = \omega_{pe}^2 + \gamma \dfrac{K_B T_e}{m_e} k_L^2$, $\omega_{pe}$ and $T_e$ have to compensate each other so that

$k_L = 2k_r$ , and frequency $\omega_L$ keeps unchanged. That is, when $T_e$ is small, $\omega_{pe}$ should be

large, namely, the observing altitude (the corresponding altitude of $\omega_{pe}$ ) is high. When $T_e$ is

large, $\omega_{pe}$ should be small, namely, the observing altitude (the corresponding altitude of $\omega_{pe}$)

is low.

3. We sincerely request the reviewer to re-consider the comment and conclusion please. If so, we

will make some modification and clarity.

Indeed, The descents of the HFPL and HFIL altitudes at EISCAT UHF, VHF and MUIR were frequently observed, which were attributed to the change in the profile of electron density [*Djuth et al.* 1994; *Kosch et al*., 2004; *Dhillon et al*., 2005; *Ashrafi et al.,* 2006; 2007] or the artificial descending layers [*Streltsov et al*., 2018].

In this paper, however, we suggested an alternative explanation for the descents of the HFPL, namely, the descents of the HFPL may be due to the enhanced electron temperature on the traveling path of the enhanced Langmuir wave rather than the change in the profile of electron density. We are trying to express that this paper should be new and meaningful.

---

## Referee Comment (RC3) · Anonymous Referee #3 · 1 May 2019

Variation in altitude of high-frequency enhanced plasma line by the pump near the 5th electron gyro-harmonic Jun Wu et al.

The paper describes the behaviour of high frequency plasma line (HFPL) during an experiment conducted near the 5th electron gyroharmonics at EISCAT heating facility on 11th March 2014. The same experimental data by these authors have already been published in a series of manuscripts (e.g. Wu et al., The extending of observing altitudes of plasma and ion line during ionospheric heating, JGR, 123, 918-930, 2018 and Wu et al., The behavior of electron density and temperature during ionospheric heating near the fifth electron gyrofrequency, JGR, 122, 1277-1295, 2017). In this paper, the

authors are discussing the effect of electron temperature on the altitude decent of the HFPL and they conclude that HFPL altitude is dependent on the dispersion behaviour of the enhanced Langmuir wave and Bragg condition, and is determined by the profiles of the electron density and enhanced electron temperature. Their discussion and conclusion adds nothing new to what has already been published and discussed by many authors. To make any firm conclusion on the altitude variation of the HFPL, you need to know the altitude that these are generated accurately, but in this paper there is no proper discussion and calculation of the reflection height and upper hybrid height for each stepping frequency based on independent measurement such as Dynasonde which can be obtained at EISCAT heating facility.

It is also important to note that when heating along the local magnetic field line, then the reflection altitude changes and is below the reflection height for vertical incidence, but can be calculated using ray tracing. It should also be noted that the linear dispersion properties of Langmuir waves in un-magnetised plasma which has been used for interpretation of the results are not appropriate and have its limitation. In summary, this paper is not suitable for publication since it does not provide any new result.

---

## Author Comment (AC3) · 6 May 2019

Reviewer 1

Thank you for the comments and suggestions. The comments given by reviewer are listed in black and ours replies are listed in red as below:

Comments to the Author

1. The paper describes the behaviour of high frequency plasma line (HFPL) during an experiment conducted near the 5th electron gyroharmonics at EISCAT heating facility on 11th March 2014. The same experimental data by these authors have already been published in a series of manuscripts (e.g. Wu et al., The extending of observing altitudes of plasma and ion line during ionospheric heating, JGR, 123, 918-930, 2018 and Wu et al., The behavior of electron density and temperature during ionospheric heating near the fifth electron gyrofrequency, JGR, 122, 1277-1295, 2017).

The fact is that from the experimental observations performed in the interval of 12.30 – 14.30 UT on 11 March 2014, we find **six** interesting phenomenon (scientific questions), as those followings,

[Figure]

**The enhanced electron temperature as a function of pump frequency**

**Fig 1.** the enhanced electron temperature as a function of pump frequency

  (1) the enhanced electron temperature as a function of pump frequency as shown in **Fig 1**. This result was reported in published paper [*Wu, J., J. Wu, M. T. Rietveld, I. Haggstrom, H. Zhao, and Z. Xu, The behavior of electron density and temperature during ionospheric heating near the fifth electron gyrofrequency, J. Geophys. Res. Space Physics, 122, doi:10.1002/2016JA023121, 2017; Wu J, J. Wu, H. Zhao and Z. Xu, Analysis of incoherent scatter during ionospheric heating near the fifth electron gyrofrequency, Plasma Sci.Technol., 19(4), doi:10.1088/2058-6272/aa58db, 2017.*].

[Figure]

**The extending enhancement in electron density**

**Fig 2.** the altitude extending enhanced electron density

(2) the altitude extending enhanced electron density as shown in **Fig 2**. This result was reported in published paper [*Wu, J., J. Wu, M. T. Rietveld, I. Haggstrom, H. Zhao, and Z. Xu, The behavior of electron density and temperature during ionospheric heating near the fifth electron gyrofrequency, J. Geophys. Res. Space Physics, 122, doi:10.1002/2016JA023121, 2017; Wu J, J. Wu, H. Zhao and Z. Xu, Analysis of incoherent scatter during ionospheric heating near the fifth electron gyrofrequency, Plasma Sci.Technol., 19(4), doi:10.1088/2058-6272/aa58db, 2017.*].

[Figure]

**The HFIL in the HB is extending in altitude (green circle), whereas the HFIL in the GB is not (yellow circle).**

**Fig 3.** a remarkable extension of observing altitudes of the HFIL

(3) a remarkable extension of observing altitudes of the HFIL as shown in **Fig 3**. This result was reported in published paper [*Wu, J., Wu J., Rietveld M. T., Haggstrom I., Xu Z., Zhao H. The extending of observing altitudes of plasma and ion lines during ionospheric heating. Journal of Geophysical Research: Space Physics, 123(1), 918-930, doi.org/10.1002/2017JA024809 2018*].

[Figure]

**The intensity of HFIL in the HB is weak (green circle), whereas the intensity of HFIL in the GB is strong (yellow circle).**

**Fig 4.** the variation in the intensity of the HFIL as a function of pump frequency

(4) the variation in the intensity of the HFIL as a function of pump frequency as shown in **Fig 4.** This result was reported in published paper [*Wu J., Wu J., Rietveld M. T., Haggstrom I., Xu Z., Zhang Y., Xu T., Zhao H. The Intensities of High Frequency-Enhanced Plasma and Ion Lines During Ionospheric Heating. JGR Space Physics. 124(1). P.603-615. doi:10.1029/2018JA025918, 2018.*]. In addition, as a phase work, we reported the original idea about the intensity of the HFIL in paper [*Altitude and intensity characteristics of parametric instability excited by an HF pump wave near the fifth electron harmonic, Plasma Sci. Technol., 19(12), 2017.* ]

[Figure]

**The HFIL altitude in the HB (green circle) is lower than that in the GB (yellow circle).**

**Fig 5.** a systematic variation in the altitude of the HFIL as a function of pump frequency

(5) a systematic variation in the altitude of the HFIL as a function of pump frequency, namely, the altitude of the HFIL in the HB is lower than that in the GB, as shown in **Fig 5.** This result was reported in published paper [*Wu, J., Rietveld, M.T., Häggström, I., Zhao, H., Xu, T. & Xu, Z. (2018). Systematic variation in observing altitude of enhanced ion line by the pump near fifth*

*gyroharmonic. Plasma Science and Technology, 20(12), 125301. https://doi.org/10.1088/2058-6272/aadd44*]. In addition, as a phase work, we reported the original idea about the intensity of the HFIL in paper [*Altitude and intensity characteristics of parametric instability excited by an HF pump wave near the fifth electron harmonic, Plasma Sci. Technol., 19(12), 2017.* ]

(6) a systematic variation in the altitude of the HFPL as a function of pump frequency, namely, the altitude of the HFPL in the HB is lower than that in the GB, as shown in **Fig 6**. However, it seems that this observation is in conflict to our usual knowledge that the HFPL altitude in the HB should be slightly higher than that in the GB due to the monotonicity of the altitude profile of ionosphere. This study result was submitted ANGEO with number angeo-2019-23.

[Figure]

The HFPL altitude in the HB (green circle) is slightly lower than that in the GB (yellow circle)

**Fig 6.** a systematic variation in the altitude of the HFPL as a function of pump frequency

As those statements above mentioned, using those observations (data) obtained in the interval of 12.30 – 14.30 UT on 11 March 2014, six interesting phenomenon (scientific questions) were studied and published respectively, implying that although the same observations (data or figures) were used in those published papers, but the focused question in those published papers is very different from each other. Thus, this manuscript (angeo-2019-23) does not repeat those study results which have been published by authors, but only repeats those radar observations. As author, however, we have the right to re-use those observations (data) in other published according to the Usage Permissions by AGU and IOP.

In this paper, we only focus on the descents of the HFPL in the HB as shown in Fig 6.

2. In this paper, the authors are discussing the effect of electron temperature on the altitude decent of the HFPL and they conclude that HFPL altitude is dependent on the dispersion behaviour of the enhanced Langmuir wave and Bragg condition, and is determined by the profiles of the electron density and enhanced electron temperature. Their discussion and conclusion adds nothing new to what has already been published and discussed by many authors.

Indeed, the descents of the HFPL and HFIL altitudes at EISCAT UHF, VHF and MUIR were frequently observed, which were only attributed to the change in the profile of electron density

[*Djuth et al.* 1994; *Kosch et al.*, 2004; *Dhillon et al.*, 2005; *Ashrafi et al.,* 2006; 2007] or the artificial descending layers [*Streltsov et al.*, 2018].

In this paper, however, we suggested an alternative explanation for the descents of the HFPL, namely, the descents of the HFPL may be ONLY due to the enhanced electron temperature on the traveling path of the enhanced Langmuir wave rather than the change in the profile of electron density.

3. To make any firm conclusion on the altitude variation of the HFPL, you need to know the altitude that these are generated accurately, but in this paper there is no proper discussion and calculation of the reflection height and upper hybrid height for each stepping frequency based on independent measurement such as Dynasonde, which can be obtained at EISCAT heating facility.

As shown in **Fig 6**, the altitude of the HFPL in the HB is lower than that in the GB. This observation is in conflict to our usual knowledge that the HFPL altitude in the HB should be slightly higher than that in the GB due to the monotonicity of the altitude profile of ionosphere.

Indeed, when the pump is stepping up and down, the reflection height and upper hybrid height will change. Due to the very small step of the pump frequency of ~ 2.804 kHz, however, the change in the reflection height and upper hybrid height of the pump should be so small that they will be covered a range gate of radar and Dynasonde and will not be distinguished accurately.

Moreover, this paper focus the observing altitude of the HFPL rather less the exciting altitude of the HFPL. For no confusion for reader, we analyze the altitude of the HFPL using a single frequency $\omega_L$ rather less a frequency range induced by the pump. Indeed, those enhanced Langmuir wave induced by the pump should share the same traveling characteristic as $\omega_L$. In this paper, we would like to describe that the pump reflection altitude in the HB should be higher than that in the GB, but the observing altitude of the HFPL in the HB is lower than that in the GB.

4. It is also important to note that when heating along the local magnetic field line, then the reflection altitude changes and is below the reflection height for vertical incidence, but can be calculated using ray tracing.

The reviewer believes that when the pump wave beam is pointing toward the geomagnetic field, the pump wave at frequency higher than 6 MHz will reflect below the upper hybrid height. Indeed, those papers (Mishin et al., JGR, 2004, V.109, A02305; Ann. Geophys., 2005, 23, p.47-53 and Fig.21 from Gurevich, 2007) share similar conclusions, but these conclusions are obtained based on the ray tracing method rather than on a more realistic plane wave assumption. In our experiment, the wave width of the pump wave is ~15 degrees. When the center of the pump wave beam points toward the geomagnetic field, a half of the pump wave beam is still above the geomagnetic field, that is, a half of the pump wave power can be above the upper hybrid height. The most important is that the PDI and OTSI were excited during the experiment on 11 Mar. 2014, as those evidences of HFIL and HFPL shown in observations, implying that the pump was not reflected near the upper hybrid altitude, but reach the parametric resonance altitude.

5. It should also be noted that the linear dispersion properties of Langmuir waves in un-magnetised plasma which has been used for interpretation of the results are not appropriate and have its limitation.

In our experiment, the UHF radar wave beam is also directed toward the geomagnetic field, and is observing the Langmuir wave and ion acoustic wave propagating along the geomagnetic field, which should not be effected by the geomagnetic field. Thus, the linear dispersion properties of Langmuir waves in un-magnetised plasma should be appropriate for our observations.

6. We sincerely request the reviewer to re-consider the comment and conclusion. If so, we will make some modification and clarity.